


Review article. **Factors leading to the occurrence of flood fatalities: a systematic review of research papers published between 2010 and 2020**

**Olga Petrucci***

*CNR-IRPI Research Institute for Geo-Hydrological Protection
Via Cavour 4-6, 87036 Rende, Cosenza
olga.petrucci@irpi.cnr.it


## Abstract

Floods kill several people every year in both developed and developing countries. The transfer of research findings from the academic community to practitioners, policy-makers and citizens may reduce the impact of floods on mortality. This systematic review analyzes 44 scientific articles extracted from WOS and SCOPUS databases written in English, published between 2010 and 2020, and focuses on flood fatalities. The first main finding of this review is the classification of drivers of flood mortality into two groups: the first group relates to the environment and the second group relates to the victims. The second main finding is the identification of strategies to practically cope with the identified drivers of flood fatalities. The main lacks of the review concern: a) the unavailability of papers based on flood fatality occurrence in developing countries and b) the absence of data focusing on people who have survived floods. This review amplifies useful findings, best practices, and lessons learned that can be useful for administrators, risk managers, and teachers of primary and secondary schools to mitigate the impact of future floods on human life.

## 1. Introduction

In July 2021, catastrophic floods affected Germany and neighboring countries, killing more than two hundred people (official data are still unavailable), an unexpectedly high number of victims for a country attentive to flood risk management. In the same month, in Henan Province, China, a 1000-year flood killed 69 people (https://floodlist.com/asia/china-henan-floods-update-july-2021). These flooding events highlighted that flood mortality is a real problem worldwide, regardless of the latitude, climate or level of development of the affected country.

In each fatal flood, one or more people died due to a series of circumstances that should be investigated in detail at the scale of a single fatality to identify which factors people need to be protected from in future floods. A reduction in *flood fatalities* (FFs) is a key objective in national and international public policies, and it requires the identification of several personal and collective risk factors as well as an understanding of their single effects and the complex relationships among them.

The literature on FFs provides a solid foundation for flood mortality reduction strategies and it can be used in educational programs which are shaped according to flood fatality drivers that are detected during floods and that occur in different environmental and societal frameworks. Nonetheless, the relevant literature on FFs is often relegated to the academic sector, and risk managers in charge of planning actions and strategies for risk mitigation are not familiar with them. Moreover, FFs are studied from different points of view by several scientific communities, and consequently, the research results are scattered in a multitude of scientific journals in disciplines ranging from emergency medicine to disaster risk reduction.

The present systematic review presents a synoptic overview of factors leading to flood fatalities. It is a practical synthesis that represents an 'operational' tool to help safeguard human life during floods. The paper is structured as follows: section 2 explains the selection criteria of the pertinent literature; section 3 presents a classification scheme for the main environmental factors driving FF; section 4 reports the flood mortality factors tied to victims; section 5 presents the possible strategies to reduce flood mortality, acting on both the environment and people; in section 5, a discussion on the limitations of this review is presented, and finally, the main conclusions are reported in section 6.

## 2. Materials and methods: criteria for the literature search

From the WOS and SCOPUS databases, papers with the following characteristics were extracted:

1.  written in English;





2.  published between 2010 and 2020 (unpublished materials, abstracts, and website articles not subjected to peer review were not included);

3.  terms contained in the title: ((flood/s) and (fatalities)); ((flood/s) and (mortality)); ((floods/s) and
(casualties)); ((flood/s) and (deaths)).

Searching key words in only the title, ensured the exclusion of papers containing generic sentences such as "floods cause several fatalities" or "flood mortality is going to increase", even if these papers do not focus on FFs. Articles selection was performed following these criteria.

• Inclusion criteria: scientific papers reporting fatal accident settings (for example, if the accident occurred outdoors or indoors, in the countryside or in an urban area) and victims' characteristics (such as their gender, age, vulnerability factors, and behaviors). Papers analyzing one or more of these factors were included in this review.

• Exclusion criteria: scientific articles on fatalities caused by tsunamis, cyclones, or coastal- and no precipitation-related flooding events (i.e., dam failures), as well as papers on fatalities caused by various concurrent hazards (i.e., landslides). Morbidity and long-term flood effects on people were outside the scope of this review.

Relevant papers were selected using a hierarchical approach (Moher et al., 2009). The search identified 93 records in WOS and 99 in SCOPUS. An additional 23 articles quoted in these selected papers were added, thus reaching a total of 215 records. After the elimination of duplicate articles, 125 records were obtained and screened (Figure 1). At this stage, 53 records were excluded because they concerned a) flood mortality
of either vegetal or animal species or b) morbidity and mortality related to long-term flood effects.

Next, 73 full-text articles were assessed for their eligibility. At this step, 29 papers were discarded because they were not pertinent to the scope of this review as they used the assessed number of FFs as a proxy for flood severity, but did not analyze either the setting of the accident or the victim's profile. Some of these
76 studies involve countries where high flood mortality makes it difficult to realize FF databases, such as India (Singh and Kumar, 2013) and Bangladesh (Paul and Mahmood, 2016). Further articles were discarded because they focused on global flood mortality in relation to economic factors (Hu et al., 2018), flood characteristics (Ahmadalipour and Moradkhani, 2019), or seasonal flood patterns (Vinet et al., 2019).

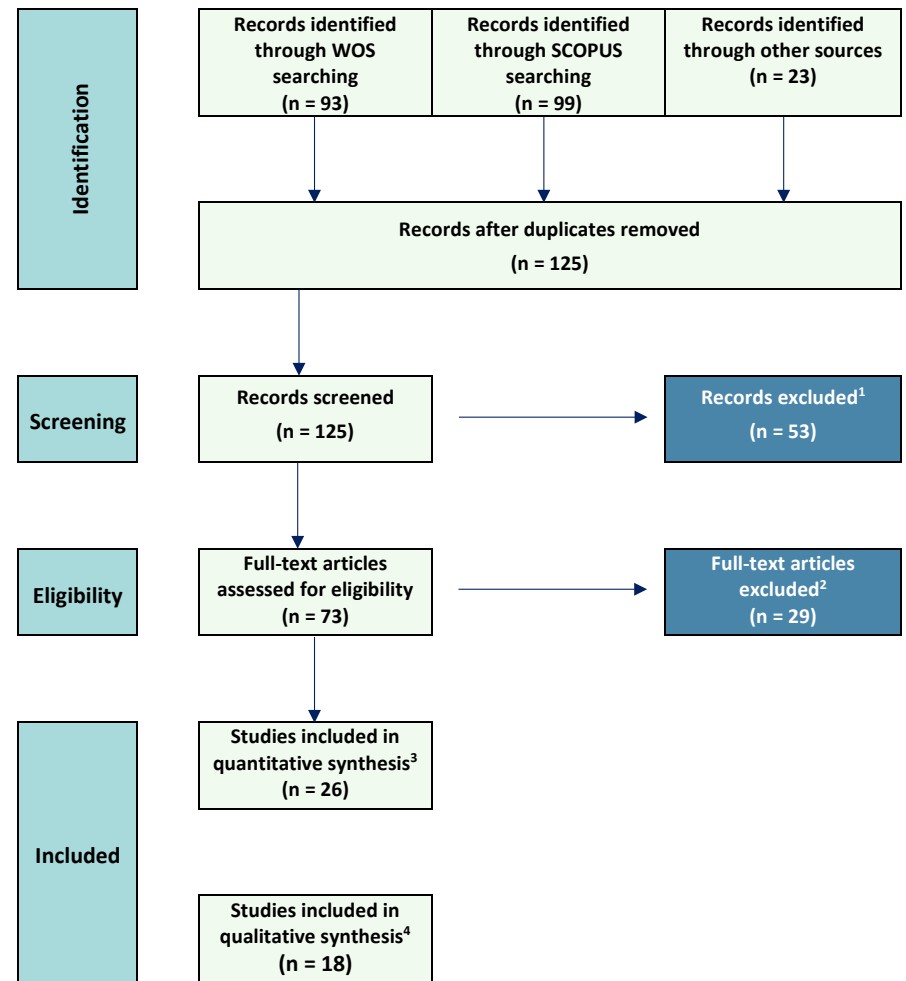

*Figure 1. PRISMA (Preferred Reporting Items for Systematic Reviews and Meta-Analysis) flow chart (Moher et al., 2009) for article selection. [1]Records excluded concerning: a) flood mortality of either vegetal or animal species or b) the long-term effects of floods on people. [2]Articles excluded because they were not relevant to the research (see the text for an explanation); [3]articles listed in Table 1; and [4]articles listed in Table 2 and quoted in the text.*

Based on the described criteria, and through full-text analysis we identified 26 papers that fully met the search criteria (Table 1), and an additional 18 papers containingcontained supportive information on

vulnerability factors and mitigation strategies (Table 2).



*Table 1. Articles selected for quantitative analysis (in chronological order of publication). Papers labeled with (\*) focus on vehicle-related accidents; DB stands for database.*

| # | Year | Authors | Title | Area | Period | Data source |
|---|------|---------|-------|------|--------|-------------|
| 1 | 2010 | Fitzgerald et al., 2010 | Flood fatalities in contemporary Australia (1997–2008) | Australia | 1997–2008 | Original DB |
| 2* | 2012 | Kellar and Schmidlin, 2012 | Vehicle-related flood deaths in the United States, 1995–2005 | USA | 1995–2005 | USA Storm Events Database |
| 3* | 2012 | Sharif et al., 2012 | Person-place-time analysis of vehicle fatalities caused by flash floods in Texas | Texas | 1959–2009 | Original DB |
| 4* | 2013 | Diakakis and Deligiannakis, 2013 | Vehicle-related flood fatalities in Greece | Greece | 1970–2010 | Original DB |
| 5 | 2014 | Špitalar et al., 2014 | Analysis of flash flood parameters and human impacts in the US from 2006 to 2012 | USA | 2006 2012 | USA Storm Events Database |
| 6 | 2015 | Becker et al., 2015 | A review of people's behavior in and around floodwater | Globe | --- | Literature |
| 7 | 2015 | Lee and Vink, 2015 | Assessing the vulnerability of different age groups regarding flood fatalities: case study in the Philippines | Philippines | 2010–2013 | Disaster Risk Reduction Council |
| 8 | 2016 | Boudou et al., 2016 | Lessons from analyzing mortality from six major flood events in France (1930–2010) | France | --- | Original DB |
| 9 | 2016 | Diakakis, 2016 | Have flood mortality qualitative characteristics changed during the last decades? The case study of Greece | Greece | 1960–2010 | Original DB |
| 10* | 2016 | Hamilton et al., 2016 | Stop there's water on the road! Identifying key beliefs guiding people's willingness to drive through flooded waterways | Australia | --- | Survey on 164 people |
| 11 | 2016 | Vinet et al., 2016 | Flashflood-related mortality in southern France: first results from a new database | France | 1988–2015 | EM-DAT, Dortmund flood DB, PRIM-NET |
| 12 | 2017 | Diakakis and Deligiannakis, 2017 | Flood fatalities in Greece: 1970–2010 | Greece | 1970–2010 | Original DB |
| 13 | 2017 | Franklin et al., 2017 | Causal Pathways of Flood Related River Drowning Deaths in Australia | Australia | 2002–2012 | Australian Coronial Info. System (NCIS) |
| 14 | 2017 | Haynes et al., 2017 | Exploring the circumstances surrounding flood fatalities in Australia-1900–2015 and the implications for policy and practice | Australia | 1900–2015 | PerilAUS, Risk Frontiers' DB |
| 15 | 2017 | Pereira et al., 2017 | Comparing flood mortality in Portugal and Greece (Western and Eastern Mediterranean) | Portugal, Greece | 1960–2010 | Original DB |
| 16 | 2017 | Terti et al., 2017 | A situation-based analysis of flash flood fatalities in the united states | USA | 1996–2014 | Literature |
| 17 | 2018 | Salvati et al., 2018 | Gender, age and circumstances analysis of flood and landslide fatalities in Italy | Italy | 1965–2014 | Original DB |
| 18* | 2019 | Gissing et al., 2019 | Influence of road characteristics on flood fatalities in Australia | Australia | 1960–2015 | Australian Coronial Inform. System (NCIS) |
| 19 | 2019 | Petrucci et al., 2019b | MEFF: The database of Mediterranean Flood Fatalities (1980 to 2015) | Mediterranean area | 1980–2015 | Original DB |
| 20 | 2019 | Petrucci et al., 2019a | Flood fatalities in Europe, 1980–2018: Variability, features, and lessons to learn | Europe | 1980–2018 | Original DB |
| 21* | 2020 | Ahmed et al., 2020 | Vehicle-related flood fatalities in Australia, 2001–2017 | Australia | 2001–2017 | PerilAUS, Risk Frontiers' DB |
| 22 | 2020 | Diakakis et al., 2020 | How different surrounding environments influence the characteristics of flash flood-mortality: The case of the 2017 extreme flood in Mandra | Greece | --- | Original DB |


| 23 | 2020 | Diakakis, 2020 | Types of behavior of flood victims around floodwaters. Correlation with situational and demographic factors | Greece | 1960–2019 | Original DB |
| 24 | 2020 | Grimalt-Gelabert et al., 2020 | Flood related mortality in a touristic island: Mallorca (Balearic Islands) 1960–2018 | Mallorca | 1960–2018 | Original DB |
| 25 | 2020 | Špitalar et al., 2020 | Analysis of Flood Fatalities-Slovenian Illustration | Slovenia | 1926–2014 | Original DB |
| 26 | 2020 | Yari et al., 2019 | Risk factors of death from flood: Findings of a systematic review | Globe | --- | Literature |

*Table 2. Articles quoted in the text as supportive material (in chronological order of publication).*

| # | Year | Authors | Title |
|---|---|---|---|
| 1 | 2010 | Yeo and Blong, 2010 | Fiji's worst natural disaster: The 1931 hurricane and flood |
| 2 | 2012 | Alderman et al., 2012 | Floods and human health: A systematic review |
| 3 | 2013 | Doocy et al., 2013 | The human impact: a historical review of events and systematic literature review |
| 4 | 2013 | Lowe et al., 2013 | Factors increasing vulnerability to health effects before, during and after floods |
| 5 | 2013 | Petrucci and Pasqua, 2013 | Rainfall-related phenomena along a road sector in Calabria (Southern Italy) |
| 6 | 2014 | Brazdova and Riha, 2014 | A simple model for the estimation of the number of fatalities due to floods in central Europe |
| 7 | 2014 | Coates et al., 2014 | Exploring 167 years of vulnerability: An examination of extreme heat events in Australia 1844–2010 |
| 8 | 2014 | Shabanikiya et al., 2014 | Behavior of crossing flood on foot, associated risk factors and estimating a predictive model |
| 9 | 2015 | Paulikas and Rahman, 2015 | A temporal assessment of flooding fatalities in Pakistan (1950–2012) |
| 10 | 2015 | Rufat et al., 2015 | Social vulnerability to floods: Review of case studies and implications for measurement |
| 11 | 2016 | Haynes et al., 2016 | An analysis of human fatalities from floods in Australia 1900–2015 |
| 12 | 2017 | Aceto et al., 2017 | Effects of damaging hydrogeological events on people throughout 15 years in a Mediterranean region |
| 13 | 2017 | Luu et al., 2017 | Analyzing flood fatalities in Vietnam using national disaster database and tree-based methods |
| 14 | 2017 | Vinet, 2017 | Flood impacts on loss of life and human health |
| 15 | 2018 | Ahmed et al., 2018 | Driving into floodwater: A systematic review of risks, behavior and mitigation |
| 16 | 2019 | Tyler et al., 2019 | A review of the community flood risk management literature in the USA: lessons for improving community resilience to floods |
| 17 | 2019 | Yari et al., 2019 | Underlying factors affecting death due to flood in Iran: A qualitative content analysis |
| 18 | 2020 | Coles and Hirschboeck, 2020 | Driving into danger: Perception and communication of flash-flood risk |

Papers in Table 1 are based mainly on original FF inventories and secondarily on literature reviews. From a geographical point of view, the studies mostly focus on Europe at the scale of either one or more countries,
followed by Australia, with studies at the scale of the entire continent. Four articles focused on the USA or states within the USA. Three studies were performed at the global scale and only one case focused on FFs in an Asian country (Table 3). In Europe, the scientific community's interest in FF appears to have increased since the second half of the study period, when 12 out of 14 studies were published. A similar trend was
detected in Australia, while for the USA, no recent papers were found that matched the selection criteria in the last three years.


*Table 3. Papers selected as the primary focus of this review by publication year and study area location*

|  | 2010 | 2011 | 2012 | 2013 | 2014 | 2015 | 2016 | 2017 | 2018 | 2019 | 2020 | Total |
|---|---|---|---|---|---|---|---|---|---|---|---|---|
| **Asia** |  |  |  |  |  | 1 |  |  |  |  |  | 1 |
| **Australia** | 1 |  |  |  |  |  | 1 | 2 |  | 1 | 1 | 6 |
| **Europe** |  |  |  | 1 |  |  | 3 | 2 | 1 | 2 | 4 | 13 |
| **USA** |  |  | 2 |  | 1 |  |  | 1 |  |  |  | 4 |
| **Globe** |  |  |  |  |  | 1 |  | 1 |  |  |  | 2 |
|  | 1 | 0 | 2 | 1 | 1 | 2 | 4 | 6 | 1 | 3 | 5 | 26 |

## 3. Environmental factors leading to flood fatality occurrence

### 3.1 Flood speed and bed load

The speed of the flood is one of the basic factors that determines a flood's impact on people, and the literature acknowledges that fast events, defined as "flash floods," are very dangerous to humans. So much so that some papers only focus on flash floods (5 papers, 19%), even if a rigorous criterion to divide flash floods from other floods does not exist. Floods in small and steep basins are classified as flash floods; nevertheless, how small and how steep the basins must be are factors that depend on the local geomorphological and climatic framework. Pereira et al. (2017a) categorized flash floods as those related to short and intense precipitation events affecting small catchment areas over a short duration.

Flash floods rapidity may surprise people, giving them a very short amount of time to decide what to do. From an emergency management point of view, the rapidity of a flash flood restricts the anticipation time of an effective response and results in less time for both warning and emergency procedure activation (i.e., road closures, rescues, and evacuations). Flash floods are very deadly in the Mediterranean regions of France (1988–2015), where half of cases took place in watersheds smaller than 150 km$^2$ (Vinet et al., 2016). In the USA (2006–2012), the shortest duration events (<1 h) caused the most fatalities (Špitalar et al., 2014), while in Texas 68% of FFs (1959–2009) were caused by flash floods (Sharif et al., 2012). Similarly, in both Greece and Portugal (1960–2010), flash floods were responsible for more than 80% of the total mortality (Pereira et al., 2017).

Non-flash floods can likewise be dangerous to human life, especially in densely populated areas and where floods are primarily caused by tropical cyclone activity and monsoon rain. Paulikas and Rahman (2015) presented a list of the 31 deadliest floods (1950–2012) in Pakistan and found that the number of FFs range from 100 (July 1959) to 10,000 (December 1965).

Another important factor is the flood bedload, which in worst case scenarios consists of huge debris, such as cars and trees, and results in trauma, orthopedic injuries, and lacerations. It is only mentioned in Špitalar et al. (2020) because of the deficiency in public data on the medical cause of death or, alternatively, by accident descriptions supplied by eyewitnesses. The impact of huge debris can cause trauma and weaken the strength of victims, promoting drownings. It is impossible to determine what happened first, trauma or drowning. A survey among people who survived floods in Iran highlighted that "When you get stuck in a flood with debris, you may have a big stick to your body, which can break your bone or graze the grass and plant roots in the goblet and you cannot breathe!" (Yari et al., 2019).

Similarly, another factor that was not analyzed in the selected literature because of the lack of data is the temperature of the floodwaters that may cause hypothermia and heart attacks, leading to death.

### 3.2 Location of the accident

Ten papers (38%) recognize that flood risk causing death is higher in rural areas due to: a) the lack of fast-responding units for rescues, evacuations, and road closures; b) low population densities which diminish the


chances of receiving first help from laypeople; c) a lack of mitigating structures, such as bridges over low
water crossings; and d) their geographic location, often at headwater basins which respond quickly to floods,
providing less warning time (Špitalar et al., 2014). In rural areas of Slovenia, the number of victims have
increased in recent years, especially in cars (Špitalar et al., 2020). A gradual shift of fatalities from urban to
rural areas has been detected in Greece (1960–2010), with an increase in vehicle-related fatalities (Diakakis
and Deligiannakis, 2017), while in urban environments, higher percentages of deaths occurred indoors,
especially among elderly people (Diakakis et al., 2020). In Australia (2002–2012), when compared to
drownings in major cities, people in remote or very remote locations were 79.6 and 229.1 times, respectively,
more likely to drown in river floods (Peden et al., 2017).

On the other hand, highly urbanized areas, if located in or near flood-prone areas, is in itself a risk, and the
increasing concentration of human activities can amplify risk factors, especially where the standard of living
is low. The standard of living in most Asian and African countries in terms of mitigation, control, and human
settlement characteristics are much lower than those in most European countries, where preparedness,
warning systems, and rescue procedures are more advanced. The strength one's house, for example, has
been demonstrated to have the greatest influence on FFs in countries such as Vietnam (1989–2015) (Luu et
al., 2017). According to some authors, flood outcomes in terms of the number of fatalities, can be only be
compared between countries characterized by similar population densities and gross domestic product per
capita (Brazdova and Riha, 2014).

Special mention must also be given to events that occurr in campsites. In these cases, the number of FFs per
event is high because people are grouped together (Aceto et al., 2017) and can be more subject to surprise,
as people vacationing in campsites in remote locations are often less weather aware and don't receive
weather warnings, limiting the speed of their response (Terti et al., 2017). In contrast, outside/open-air or
close-to-stream events more typically impact individuals than groups of people (Terti et al., 2017). A gap in
knowledge exists about the occurrence of accidents related to mobile homes, for which no mention has been
found in the analyzed literature, even though they are expected to happen.

### 3.3 Vehicle-related flood fatalities

Vehicle-related accidents are frequent and are often analyzed in the literature, to the point where some
papers restrict their analysis to them (13 articles, 50%). Since 1960, an increasing number of cars has
contributed not only to people's mobility but also to their augmented exposure to floods (Petrucci and
Pasqua, 2013; Sharif et al., 2012). In the USA, more than half of flash flood fatalities were vehicle-related,
and males died at twice the rate of females for those aged 40 years old and older (Kellar and Schmidlin, 2012).
With regard to their location within the vehicle, in Greece (1970–2010), victims were equally divided between
drivers and passengers. Males showed an increased representation among victims (70%) and among driver
victims (86%), while passenger victims were found to be almost equally divided between males and females
(Diakakis and Deligiannakis, 2017). In Australia (2001–2017), 60% of vehicle-related FFs were drivers, 31%
were passengers and 8% were unknown. Females represented 43% of victim passengers and only 28% of
driver fatalities. Drivers were mainly over 30 years old (88%), while as might be expected, among passengers,
children (46%) were more numerous than adults. For drivers, middle-aged and elderly males were
overrepresented. As passengers, young women and children seemed more vulnerable (Ahmed et al., 2020).
Gissing et al. (2019) identified typical features in vehicle-related locations in Australia: a) small upstream
catchment with rapid rising of floodwaters; b) an absence of roadside barricades; c) deep flood waters
immediately adjacent to the roadway; d) an absence of lighting; e) dipping road grades that lead to
floodwaters and increase once a vehicle enters them; and f) the lack of curbs or ditches. The main
mechanisms for vehicle-related FFs are a) a vehicle on a flooded roadway may result in a stalled engine,





stranding the vehicle and its occupants as the flood waters rise; b) moving water may push a vehicle off the roadway into deeper water, where it becomes submerged; c) persons may be ejected from a vehicle caught in floodwaters, or they may attempt to escape the flooded vehicle and once in the floodwaters, they may drown or be killed by trauma if they cannot escape; or d) victims may exit the flooded vehicle and attempt to escape on foot, and then fall or stumble into deeper water from which they cannot escape (Kellar and

Schmidlin, 2012).

### 3.4 The timing of FF accidents

Except in cases where someone witnessed the accident, the timing of the accident is often largely approximated. The hour at which the accident occurred determines the visibility, which is an important factor

affecting, i.e., a driver's capability to judge the depth and speed of flowing water. Moreover, in addition to the amount of daylight, visibility depends on other factors that could worsen it, such as intense rain, hail, and fog. For this reason, time alone is not fully indicative of the visibility and cannot be used as a predictor in the probability of fatal accidents; in fact, conclusions on this factor are discordant.

In three papers (12%), accidents were reported to mainly occur at night. In the USA, 51% of flash flood fatalities (2006–2012) occurred between 22:00 and 05:00, probably because flooded roadways cannot be seen as easily and people drove inadvertently into dangerous conditions (Špitalar et al., 2014). In Greece (1970–2010), fatal incidents occurred mostly during nighttime, outdoors and in rural areas of the country

(Diakakis and Deligiannakis, 2017). In contrast, in Australia (1900–2015), the highest proportion of FFs (36.6%) occurred in daylight, and the people were probably aware of the flood risk, but were surprised by the depth and/or speed of the flowing water (Haynes et al., 2016).

Some papers also analyzed the day of the week to look for relationships among the peoples' habits and

204 intersections with their daily activities and day of accident occurrence. In Australia (2001–2017), vehicle-related FFs occurred mostly just before the weekend (Friday, 24%), in many cases at night, and noticeably on weekends (Sunday, 17%; and Saturday, 15%), while on working days, large numbers of incidents occurred on Mondays (14%) (Ahmed et al., 2020).

## 4. Factors related to victims

### 4.1 The gender of fatalities

The gender of the victim acts differently according to the socioeconomic characteristics of their community. Studies performed in Europe, the USA and Australia revealed a constant higher number of male fatalities, as

reported in 65% of the papers in Table 1. In the USA (1996–2014), males exceeded females in all circumstances (Terti et al., 2017). In some European countries (1980–2018), there were fewer female victims than males, except for elderly victims (Petrucci et al., 2019a). In Australia (1900–2015), 79.1% of fatalities were males, even if since the 1960s, the proportion of female to male FFs has increased, and for both

genders, the proportion of fatalities decreases steadily with age (Haynes et al., 2016). In Texas (1959–2009), significantly more males died in vehicle-related accidents than females (Sharif et al., 2012a). Males' vulnerability can be related to their stronger exposure to floods and due to the higher proportion of males that drive vehicles because of either their wider mobility or their outdoor work. In contrast, female fatalities

become more numerous in underprivileged frameworks, as in some Asian countries, even if those sectors of the globe are characterized by knowledge gaps in FFs, due to the lack of inventories reporting the gender of FFs.


## 4.2 The age of fatalities

Findings on the age of FFs are rather complex to summarize and compare because of a) the different age ranges used in each study and b) the analysis of age is often combined with other characteristics (i.e., gender) or specific circumstances (i.e., vehicle-related fatalities).

In European countries (1980–2018), the majority of FFs were between 30 and 64 years, thus in their most
productive working years, which explains why several FFs occurred outdoors while heading from home to work premises or vice versa. In contrast, elderly (retired) people were more frequently affected indoors at home (Petrucci et al., 2019a). Even in Australia (1900–2015), elderly people were more frequently trapped by floods in their homes, while younger adults and their children were dragged outdoors, but 43.4% of deaths
occurred in those younger than 29 years (Haynes et al., 2016). In the USA (1996–2014), almost 50% of the males in vehicle-related fatalities were between 24 and 62 years old, according to the fact that 80% of licensed drivers are between 20 and 64 years old (Terti et al., 2017).

According to Kellar and Schmidlin (2012), the death rate for both genders at the youngest age range (under
5 years) is more than 2.5 times the death rate of those aged 5–14 years old because of their dependence on others to rescue them from floodwaters. Furthermore, these authors identified higher death rates in males over 70, due to their greater vulnerability to trauma, which decreased their mobility and ability to escape.

## 4.3 The residency of the fatalities

It can be argued that people living in a certain location have a deeper understanding of their environment compared to persons who are not residents, i.e., tourists. Residents act more according to their knowledge of the place they live than to following safety procedures: environmental familiarity may cause individuals to underestimate risks making people more likely to voluntarily enter floodwaters. Accordingly, in some cases,
the large majority of victims were residents in the proximity of the accident location, and victims with a visitor status were not found to have an increased drowning risk, such as in Australia (2002–2012) (Peden et al., 2017). In Greece (1970–2010), the majority of driver fatalities (78.05%) permanently lived less than 20 km from the location of the accident (Diakakis and Deligiannakis, 2017).

In contrast, due to their unfamiliarity with the local environment, tourists may be figured as more vulnerable because floods are more likely to surprise them, and they may have difficulties accessing whether the event will affect them and how they should respond. Nevertheless, they may be less vulnerable than residents since they have no emotional attachment to the location and therefore may be able to make decisions more easily
(Becker et al., 2015).

## 4.4 Victims' vulnerabilities

Individuals affected by **physical and/or psychological disabilities** are clearly vulnerable, as these conditions can lead to impaired responses, reducing the chance of survival with respect to people not experiencing any
vulnerability factors. This was tragically experienced in the flood that in 2000 where 13 people were killed in a campsite in southern Italy, five of whom were mobility-impaired people (Aceto et al., 2017). Nevertheless, victims' vulnerabilities are rarely reported in FF inventories because of privacy laws.

Many other issues could act as vulnerability factors, such as the weight and height of individuals, their
mobility in water, their clothing and footwear, and the use of support or carrying of loads (Brazdova and Riha, 2014). In low-income countries, **being a woman** can represent a vulnerability factor, according to the higher number of victims among females (Doocy et al., 2013). In those societies, females, especially very young and elderly individuals, tended to be at higher risk. In contrast, in high-income communities, males and older
persons are more vulnerable (Alderman et al. 2012). In South Asia, female vulnerability is related to the following factors: 1) women are more likely to stay at home rather than evacuate to a shelter; 2) wearing a



dress restricts their movements; 3) cultural shame prevents women from escaping to public areas if their clothing is ripped; 4) their inability to swim, which is a consequence of cultural norms; and (5) being less well
nourished (Yeo and Blong, 2010).

**Poverty and low education** can be considered intrinsic risk factors for flood mortality (Yari et al., 2020). In low-income countries, as a result of the relatively high rates of poverty and inadequate mitigation measures, people are more vulnerable to floods. According to Lee and Vink (2015), FFs correlates with several factors:
a) people's marginality relevant to demographics, disability, and other socioeconomic weaknesses; b) illegal settlement, urban sprawl, and other environmental deterioration; c) political corruption related to accountability, effectiveness, and regulatory quality; and d) coping capacity, including disaster education and community programs.

Member countries of the OECD (Organization for Economic Co-Operation and Development) are relatively comparable, unlike low-income countries, because the impacts of floods change according to the percentage of the gross domestic product spent on flood management (Lowe et al., 2013). Moreover, even within medium- and high-income countries, disparities place vulnerable groups, as, i.e., poor communities of color,
ethnic minorities, urban homeless, and people with chronic diseases, at severest flood risk (Alderman et al., 2012).

In some coutries, **cultural features** may also affect people's behavior to floods; in Pakistan, there is the popular belief that little can be done to mitigate catastrophic events because they are caused by God's wrath
(Paulikas and Rahman, 2015).

Finally, impaired responses can be related to temporary vulnerability conditions, as in the case of **alcohol or drug use**. In Australia (2002–2012), common causal factors for falls into flooded rivers include being alone and having a blood alcohol content ≥0.05% (for adults) (Franklin et al., 2017). Moreover, in vehicle-related
FFs (2001–2017), for the 38 cases in which drug and alcohol levels were tested, 55% of the FFs were identified as having alcohol in their urine or blood from autopsy reports (Ahmed et al., 2020).

### 4.5 Victims' hazardous behaviors

The frequent mention in the literature of risk-taking actions clearly indicates that human behavior is a crucial
factor in flood mortality, and it must be fully understood to focus on customized mitigation strategies such as educational campaigns. Regarding gender, men seem more prone to exceed the standard safety rules, take more risks and put themselves in danger, e.g., to rescue people, pets, or belongings.

Males between 10 and 29 years old or those over 60 enter floodwaters for five reasons: (a) for recreational
activities; (b) to reach a destination; (c) to retrieve property, livestock, or pets; (d) to undertake employment related duties; and (e) to rescue someone (Becker et al., 2015). Nevertheless, crossing a bridge, causeway, or culvert is a frequent behavior detected in both males and females (Haynes et al., 2016).

The motivation to enter floodwaters is related to both incorrect risk perception (i.e., underestimating the
flood risk or being unaware of it) and social influences (i.e., following others' actions and social pressures) (Pereira et al., 2017). In the case of drivers, the decision to drive into or turn back from floodwaters is the result of their risk perception combined with other factors (e.g., individual, social, environmental, etc.) (Ahmed et al., 2018), and even in cases where people are aware of the risks, the depth and/or the speed of
the water can take them by surprise due to an underestimation of the water depth or flow velocity (Coates et al., 2014).

**Trying to save belongings** is one of the main motivations that pushes people to behave recklessly. An example is reported by Vinet (2017) in the Côte d'Azur (France) in 2015, when eight people died as they tried
to get their cars out of underground parks. Most likely, they did this after the flood warning had been issued and had managed to save their cars on several previous occasions, but lost their lives during the last attempt.



People between 18 and 35 years old seem prone **to cross floods on foot** because (a) they do not take flood warnings seriously, (b) they believe they are good swimmers, and (c) they do not have previous experience with floods (Shabanikiya et al. 2014; Yari et al., 2019). In contrast, it appears that individuals who know how to avoid floods (i.e., through personal experiences or by asking others for advice), are less likely to enter floodwaters (Coles and Hirschboeck, 2020).

**Driving on a flooded road** is dangerous because water flow/velocity can influence accidents even when the water level is only 20 cm (Ahmed et al., 2020). Accidents mostly occurred close to home, where drivers are expected to be familiar with their surroundings and the nature of the roads. A false sense of security can develop when an individual is inside a vehicle, and this sense of security may be increased by a driver's overconfidence in his or her ability to drive, past successful crossing experiences, and overconfidence in the vehicle type. In Australia (2000–2015), an increase in 4 WD vehicle drivers' death was detected (Franklin et al., 2017): a reliance on thinking that large vehicles, such as SUVs, are sturdy may increase hazardous behaviors (Becker et al., 2015). Even if drivers may identify the potential risk, they fail to personalize it, believing that it does not apply to them (Gissing et al., 2016; Pearson and Hamilton, 2014), making them impatient and thinking that they are invincible (Franklin et al., 2014). Moreover, some car commercials exhibit drivers climbing mountains and crossing rivers, presenting SUVs as invincible cars, and this can promote reckless behavior in people watching these commercials, who do not realize that the ads are shot on a cinematographic set, and thus are different from real life.

Alcohol and drugs can contribute to an incorrect perception of danger: they can increase drivers' hazardous behavior (i.e., ignoring warnings and indicators) or can slow down decision making reaction times and their capacity to move, swim, or escape from the vehicle, and/or help others escape (Ahmed et al., 2020). In the majority of cases, drivers chose to enter flooded areas, either to travel across, to save someone, or to recover something (Diakakis and Deligiannakis, 2017).

The literature described **flood tourism** as a series of misconduct and reckless behaviors undertaken by people on either bridges or banks of rivers to watch floods: they underestimate the dangers of the flood and often end up being swept away by the water. People can behave dangerously when their curiosity pushes them to see the flood as closely as possible and observe what is happening. These behaviors are more frequently detected in children and teenagers who are walking or playing in the floodwaters (Terti et al., 2017), and young people (0–19 years) seem to account for the greatest proportion of FFs engaged in activities near floodwaters (Haynes et al., 2016), often taking selfies or videos in extreme situations. The number of people entering floodwaters for recreational purposes is increasing due to the spirit of emulation created by videos posted on social media (i.e., people driving through flooded roads). In central Europe, recreational boaters have attempted to boat or raft on flood waters which have resulted in crashes, capsizing and drownings in high-velocity streams (Brazdova and Riha, 2014).

People working in **specific occupations**, i.e., emergency services, utility maintenance workers, mail delivery personnel, and miners may be more likely to enter floodwaters (Becker et al., 2015). Victims related to work activities are not numerous (i.e., in Australia, 5% of fatalities occurred to individuals who were performing work-related duties at the time of the fatal incident) (Ahmed et al., 2020). They are more frequently males because males are still more likely to work outdoors than females, and until recently, they also outnumbered females in rescue services (e.g., firefighters, police, and defense forces) (Salvati et al., 2018)

## 5. Possible strategies

The literature suggests general recommendations for reducing both flood risks and flood mortalities. The general agreement around flood mortality is that it is the result of a series of concurrent factors and,



therefore, only strategies that are based on multiple approaches can produce significant effects. The specific countermeasures identified in the selected literature are summarized in the following subsections.

## 5.1 Actions related to environmental factors

One of the possible strategies to reduce flood mortality is to act on environmental factors. In the case of flash floods, for example, the implementation of more accurate **forecasts and warning systems**, based on forecasted rainfall amounts and their effects on small watersheds, could lead to advanced forecasts, thus expanding the available time to issue warnings and perform protective actions, i.e., roads closures.

More complex systems have been suggested to avoid vehicle-related accidents, such as flood depth indicators and **warning signage**, with the installation of visual cues (flash lighting and automated barricades) in high-risk locations (Ahmed et al., 2020), especially at night, when drivers may not be able to judge the water depth, current, or their location on the roadway (Kellar and Schmidlin, 2012). Checking the

effectiveness of existing risk indicators and regularly evaluating their efficacy with road users should ensure their efficiency (Ahmed et al., 2020). In underground parking garages, signs alerting drivers of the risks of inundation in the case of floods can be useful to avoid risky behaviors in people trying to save their cars.

## 5.2 Actions related to people's risk awareness

A series of actions can improve people's awareness and promote their self-protective behaviors can be collected, such as memorial actions of past floods and dissemination of information about recent floods and flood risks in general. As it is impossible to put warning signs everywhere along river networks, the safety of nearby citizens can only increase if they are aware that during floods they need to be cautious and **reduce**

**their risky behavior** by limiting their displacement and avoiding all the areas adjacent to the river network.

A frequent proposal is the implementation of long-term customized **educational strategies** communicating the risk of floodwaters to children prior to their teenage years and highlighting the dangers of driving through floodwaters (Peden et al., 2017). Becker et al. (2015) report examples of public education programs issued

in the USA and Australia aiming to deter people from entering floodwaters; for example, suggesting that children should not play in areas that are exposed to flood risks. Educational campaigns should involve local councils, schools, emergency service agencies, police, vehicle manufacturers, and insurance companies to reinforce messages about the potential dangers of entering floodwaters (Ahmed et al., 2020). Moreover,

instructions must be disseminated to driving schools and they should systematically teach future drivers how to manage flood risks in the event of a vehicle being trapped in water.

To address the cases of people working in specific occupations, i.e., emergency services and rescue personnel, workplace health and safety initiatives could assist with education about the dangers of

floodwater entry, the development of skills to assist with appropriate decision-making when faced with floodwaters, physical training for floodwater situations, and the provision of appropriate safety equipment if floodwater entry is required as part of their work (Becker et al., 2015).

Other authors suggest **improving communication during the events**, i.e., targeting people in rural and

remote areas with prevention messages (Peden et al., 2017), and social media could play an important role. In addition to hazardous behaviors, people can also behave in a self-protective way during floods, successfully managing to save themselves, and these cases should be accurately analyzed to identify and promote safe behaviors. Unfortunately, data on people who were able to save themselves from a flood are very difficult

to collect because they are reported in online news and newspapers as generic and in concise descriptions (i.e., "Several people saved themselves waiting for rescuers on the roof of their house"). Nevertheless, some self-protective behaviors, such as getting onto the roof/upper floors, climbing trees or climbing on top of a car roof, have been highlighted in people involved in floods who have survived (Aceto et al., 2017), even if


much more data on this topic are necessary to discern if a particular behavior is actually safe or if the people
were simply lucky that they were able to save their life.

## 6. Discussion, limitations and knowledge gaps

The studies reviewed provide an updated overview of recent research on FFs. Nevertheless, some knowledge

gaps still exist, and their possible effects on the quality of this review are described.

- **Geographical distribution of research**. There is a lack of information in less developed countries that
are strongly and frequently affected by mortal floods. In contrast, in developed countries, data and
research are widely available, even if their use in the practice of flood risk management by national and

local agencies is not demonstrated. The majority of studies center on Europe and Australia, while studies
concerning other continents/countries, such as Asia and South America, were not identified.
Researchers working on flood risk in those areas should focus on flood mortality, collecting data that are
currently only largely available from web sources (i.e., national news agencies and local communities of

meteorological amateurs) and social media (i.e., eyewitness descriptions).

- **Reliability of flood fatality data**. Several articles are based on data from global disaster databases such
as NATHAN (Natural Hazards Assessment Network) of the reinsurance company Munich Re or the EM-
DAT (Emergency Events Database) from the Centre for Research on the Epidemiology of Disasters of the

Université Catholique de Louvain. These databases focus on major disasters, and they only report the
number of FFs, without specific data on the victims and dynamics of the accidents. They are often used
in papers on flood mortality in developing countries, but these articles were excluded due to the absence
of data on both the environment and the victims.

- **Data numerousness**. Some of the factors described in section 3 are reported in a few of the analyzed
papers. For example, the effect of bedload can be reasonably assessed as a very important factor;
nevertheless, due to the lack of data on the medical cause of death, few authors discuss it. Section 3
attempts to systematize, in a panoramic view, all the flood mortality factors described in the analyzed

literature, regardless of the number of papers acknowledging their importance.

- **Possible bias due to analysis of specific accidents**. Because of the high frequency of vehicle-related FFs,
several papers exclusively deal with them, and thus, these papers supply only a partial view of flood
mortality, neglecting the other kinds of accident dynamics. This must be considered in order to correctly

assess results on factors such as gender and the age of victims, taking into account that in those papers,
the results were obtained only on a subsample of the victims, and include only those who died in vehicle-
related accidents.

In addition to the described problems, the present review has some advantageous qualities:

- *Newness*. This review highlights the most recent trends in people's behavior and then supplies an
updated framework of factors currently leading to FF occurrences, considering the ongoing changes in
people's lifestyles. With the advent of social media in recent years, for example, reckless people can
post videos of themselves on social media of hazardous behaviors in or near floods, increasing the spirit

of emulation of people watching those videos.

- *Usefulness*. This paper presents the main factors of FFs and provides suggestions to minimize them
based on innovative and practical lessons that could help decision-makers better manage flood risks in
communities. It also supplies a guide to truly cope with the occurrence of FFs and to help build a

roadmap towards efficiently reducing the effects of each factor, thus answering the call for a simple
transfer of the important findings from the academic community to practitioners and policy-makers.



- *Perspective*. This paper also focuses on the knowledge gaps still existing that researchers all around the world can fill to reduce flood mortality and improve communities' resilience to future floods.
Investigating flood mortality in developing countries and determining successful protective behaviors carried out by people whom survive a flood is a research topic that requires additional work. This review also highlighted the two-fold role of social media. On the one hand, social media can be a useful data source and an effective means for providing flood advice and spreading educational messages to specific
age classes, such as young people. On the other hand, they can be a mouthpiece for disseminating bad examples of people's behaviors during floods that could be emulated by others.

## 7. Concluding remarks

Ten years of literature on flood mortality has highlighted that situational rather than generic examination of
448 fatal accidents is required to realistically capture risky behavior during flood events (Terti et al., 2017). The numerous factors leading to flood fatalities are highlighted in this systematic review and have been sorted into two groups, one group relating to environmental factors and another group associated with the victims, even if sometimes a clear distinction cannot be made due to the interrelations between all the relevant
factors. Accordingly, strategies to reduce flood mortality have also been divided into those that try to modify environmental factors and others that try to improve the risk awareness of citizens. The complexity of the problem suggests the simultaneous implementation of both kinds of remedials synergically, to help combat flood mortality.

The research highlighted two important data gaps. The first concerns flood mortality in developing countries, where detailed inventories of flood fatalities should be performed, even for short periods (5 to 10 years). The second gap concerns the actions performed by people who managed to save themselves during a flood. Future research should focus on these data gaps, going deeper into the dynamics of single accidents and
researchers should collect more data on the interaction between individuals and flood events. In addition, the details of recent floods in which survivors can tell their story should be studied. In Europe, Australia and the USA, flood fatality inventories are available, and consequently, the role of factors such as the gender and the age of victims is known. In contrast, developing countries require detailed flood fatality inventories,
instead of the simple counts of victims per flood event. Developed countries, require studies to shed light on accident circumstances to reach a better understanding of the relative importance of drivers of flood mortality and how they can be used to design mitigation measures.



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
