# Peer review of "Review article. Factors leading to the occurrence of flood fatalities: a systematic review of research papers published between 2010 and 2020 Olga Petrucci"

_Natural Hazards and Earth System Sciences, 2021_

## Referee Comment (RC1)

**Notes on petrucci et al. : factors leading to the occurrence of flood fatalities….**

**1- General comments**

This paper is very useful as it addresses a more and more relevant issue for risk managers and authorities. The methods and the criteria used to analyze the papers are well described and clearly exposed. The general schedule of the paper is correct.

**2- Specific comments**

The risk of Dike break is not addressed. We agree that Dam break is a specific issue dike break is part of the flood scenarios and is a frequent factor in flood death toll. (see New Orleans, Germany in July 2021; southern France in 1999 and 2002 or Spain. A little paragraph on the literature related with dike break and its consequences on mortality would be welcome especially for the epidemiologic point of view (night, surprise effect…)

L. 220 Male mortality is also due to work in rescue services (linked § 4.5 l. 293-298). It may be better to move the paragraph on male behavior (293-298) to § 4.1.

l. 409 :on the reliability of data, the article of Altez and Revet on the Vargas event in Venezuela is an interesting event to show the frailty of global toll on mortality. They show that the death toll usually admitted for the Vargas event had been largely over estimated.

Altez, R., Revet, S., (2005). *Contar los muertos para contar la muerte : discusion en torno al numero de fallecidos en la tragedia de 1999 en el estado Vargas – Venezuela.* Revista Geografica Venezolana, Numero especial 2005, pp. 21-43

In § 5.1 : For the proposition of action to reduce human toll, a paragraph on building retrofitting or/and adaptation to save life (roof evacuation, balcony or terrace, addition of a storey) would be welcome. Many papers point out the need to improve building adaptation to reduce the number of fatalities.

**3- technical comments**

l. 13 : Abstract : replace several by numerous

Line 47 -48 two section 5

L. 285 and L. 328 the problem of Alcohol and drugs is evoked twice. May be better to put together both paragraph on this problem.

Line 87 contained

---

## Author Comment (AC1)

**1- General comments**

This paper is very useful as it addresses a more and more relevant issue for risk managers and authorities. The methods and the criteria used to analyze the papers are well described and clearly exposed. The general schedule of the paper is correct.

Many thanks for these comments on the work

**2- Specific comments**

The risk of Dike break is not addressed. We agree that Dam break is a specific issue dike break is part of the flood scenarios and is a frequent factor in flood death toll. (see New Orleans, Germany in July 2021; southern France in 1999 and 2002 or Spain. A little paragraph on the literature related with dike break and its consequences on mortality would be welcome especially for the epidemiologic point of view (night, surprise effect…)

Actually, dike break was out of the purposes of the review, as described in the section 2. *Materials and methods: criteria for the literature search, Exclusion criteria*, line 65. Then, to introduce this inundation mechanism I should change the selection criteria. Moreover, in this section, I can introduce an explanation of the factors driving to this exclusion in a note like this:

*\* due to the character of major disaster, dam failures were excluded because literature generally analyze dam structure/height/building material, failure mechanism, volume of water released etc. and their impact on the whole of population exposed, not on single persons.*

L. 220 Male mortality is also due to work in rescue services (linked § 4.5 l. 293-298). It may be better to move the paragraph on male behavior (293-298) to § 4.1.

Yes, maybe it is more appropriate, it will be changed

l. 409 :on the reliability of data, the article of Altez and Revet on the Vargas event in Venezuela is an interesting event to show the frailty of global toll on mortality. They show that the death toll usually admitted for the Vargas event had been largely over estimated.

Altez, R., Revet, S., (2005). *Contar los muertos para contar la muerte : discusion en torno al numero de fallecidos en la tragedia de 1999 en el estado Vargas – Venezuela.* Revista Geografica Venezolana, Numero especial 2005, pp. 21-43

Thank you for the suggestion, I can quote this paper as an example of inaccuracy as follow:

*In some countries, and especially for larger disasters, the uncertainty on the number of fatalities still remain after the end of the event. This is the case of the disaster suffered by Vargas state (Venezuela, December 1999), in which medias, political authorities, national and international aids "soon had estimated numbers that quickly became inaccurate and lifted its limits to tens of thousands dead persons, without specifying the totals" (Altez and Revet, 2005).*

In § 5.1: For the proposition of action to reduce human toll, a paragraph on building retrofitting or/and adaptation to save life (roof evacuation, balcony or terrace, addition of a storey) would be welcome. Many papers point out the need to improve building adaptation to reduce the number of fatalities.

Thank for this suggestion. I can introduce the following:

*As far as the measures to avoid loss of life inside buildings, a complete analysis of the topic is presented in (NYCPlanning, 2014), where four steps facilitate the informed decision-making and the correct planning of strategy to protect buildings and people living inside them (1. identify your flood risk; 2. identify your flood elevation; 3. review relevant regulations; and 4. identify your adaptation strategy). In detail, six major retrofitting methods are proposed: elevation, relocation, demolition, wet*

*floodproofing, dry floodproofing, and barrier systems. For each of these strategies pros/cons analysis is presented, in terms of economic cost, loss of usable areas inside buildings, effects on insurance premiums, and actual protective effectiveness. A wide portfolio of case studies illustrates in detail the retrofit strategies for several different types of houses.*

**3- technical comments**

l. 13 : Abstract : replace several by numerous

Thank you, modified

Line 47 -48 two section 5

Thank you, modified

L. 285 and L. 328 the problem of Alcohol and drugs is evoked twice. May be better to put together both paragraph on this problem.

Yes, modified

Line 87 contained

Thank you, modified

---

## Author Comment (AC2)

**Comment on nhess-2021-269**
Anonymous Referee #2

The paper deals with analysis of factors leading to the occurrence of flood fatalities based on review of selected number of per-reviewed research papers published between 2010 and 2020. With respect to great attention devoted to the study of different aspects of flood fatalities it represents an actual topic. The paper has a good potential to be published in Natural Hazards and Earth System Sciences.
I thank Anonymous Referee #2 for this comment

I recommend to the author to take in account some points reported below.
General comments:

- Please reconsider the use of terms "fatality" and "victim". Although fatality is a clear term concentrated only on the death, under victim can be included fatalities and injured what is probably not intention of this article.
  I agree with this comment and I can review the entire paper in order to avoid the term "victim" that can be misunderstood.

- Lines 25-26: Related data to July 2021 in Germany are still not available? Any quotation is needed there.
  Currently data are available in a Wikipedia page based on local newspapers of the countries affected, and reporting the total number of fatalities. I can quote it (https://en.wikipedia.org/wiki/2021_European_floods).

- Lines 344-349: I have doubts about including "specific occupation" in hazardous behaviour. Please reconsider.
  You are right, it seems not appropriate. I can move this sentence to the section concerning vulnerabilities. In fact, people doing some specific occupations are more vulnerable because they are exposed to dangerous situations more frequently than other people (due to their work obligations).

- I recommend to finish Concluding remarks by anything in sense as I tried to express here: "Let us hope that this review showing manifold aspects of the study of flood fatalities will stimulate further research on this field with respect its other aspects, data mining in regions where such data have been not collected yet or where still exists potential for complementing already existing datasets."
  Thank you for this suggestion, I think it can be useful to add a sentence like this to address further research, and put the light on the work that still should be done in the different countries.

- I would like to see anywhere some sentence that selection of papers for the overview does not necessarily cover all spectrum of related FF papers, because FFs can be analysed in the structure followed in this paper also in non-considered papers, e.g. together with other weather-associated fatalities.
  I could add a sentence in Section 2, Materials and methods to explain that:

  *"It must be taken into account that the overview does not necessarily cover all spectrum of related FF papers, but only the papers fitting the selection criteria used in this review (papers published outside the period from 2010 to 2020, i.e., are excluded).*

Specific comments:
Please check if in below quoted cases is the use of related words correct:

Line 43: is it possible to use other form than "a synoptic overview" giving some connotation to meteorology ("synoptic")?

Yes, it could be changed in "a panoramic framework"

Line 61: rather "and fatality characteristics". Line 87: "containing contained"?

Yes, it can be changed

Line 113: The rapid onset of flash floods … a very short time …

Yes, it can be changed

Line 120: what "the total mortality" you have in mind? total flood mortality?

Yes, *total flood mortality*. It is a mistake, I forgot to specify it.

Line 154: can be only be compared?

It is a mistake: *can only be compared*

Lines 170-172: fatalities instead of victims?

Yes, it must be changed

Line 220: sectors of the globe or rather regions of the globe?

Yes, it can be changed

Line 246: (78%) instead of (78.05%).

Yes, it can be changed

Lines 320-321: abbreviations (WD, SUV) should be explained.

Yes, it can be changed

Lines 353-354: The sentence "The specific …" is not needed.

I can eliminate it

Line 358: better rainfall totals than rainfall amounts.

Yes, it can be changed

Line 402: mortal flood?

It is a mistake: *fatal flood*

Lines 406-408: what about some official data of national institutions and bodies (ministry, policy, statistical services etc.)?

Yes, these data source can be added. Nevertheless, social media and websites can contain details that institutions often do not collect (also in developed countries), as gender and age of FF, or accident descriptions.

Line 427: advantageous qualities?

I can change in "*The novelty of the study is in the following points*"

Line 428: newness?

I can change in "*novelty*" or in "*originality*"